# Pathogenesis and Current Treatment Strategies of Hepatocellular Carcinoma

**DOI:** 10.3390/biomedicines10123202

**Published:** 2022-12-09

**Authors:** Deniz Tümen, Philipp Heumann, Karsten Gülow, Cagla-Nur Demirci, Lidia-Sabina Cosma, Martina Müller, Arne Kandulski

**Affiliations:** Department of Internal Medicine I, Gastroenterology, Hepatology, Endocrinology, Rheumatology and Infectious Diseases University Hospital Regensburg Franz-Josef-Strauß-Allee 11, 93053 Regensburg, Germany

**Keywords:** HCC, NAFLD, NASH, non-alcoholic fatty liver disease, non-alcoholic steatohepatitis, lenvatinib, atezolizumab, pembrolizumab, trans-arterial chemoembolization (TACE), immune checkpoint inhibition, PDL1, CTLA4, P53, TERT, Barcelona Clinic Liver Cancer classification system (BCLC), Tyrosine kinase inhibition (TKI)

## Abstract

Hepatocellular carcinoma (HCC) is the most frequent liver cancer with high lethality and low five-year survival rates leading to a substantial worldwide burden for healthcare systems. HCC initiation and progression are favored by different etiological risk factors including hepatitis B virus (HBV) and hepatitis C virus (HCV) infection, non-/and alcoholic fatty liver disease (N/AFLD), and tobacco smoking. In molecular pathogenesis, endogenous alteration in genetics (*TP53*, *TERT*, *CTNNB1*, etc.), epigenetics (DNA-methylation, miRNA, lncRNA, etc.), and dysregulation of key signaling pathways (Wnt/β-catenin, JAK/STAT, etc.) strongly contribute to the development of HCC. The multitude and complexity of different pathomechanisms also reflect the difficulties in tailored medical therapy of HCC. Treatment options for HCC are strictly dependent on tumor staging and liver function, which are structured by the updated Barcelona Clinic Liver Cancer classification system. Surgical resection, local ablative techniques, and liver transplantation are valid and curative therapeutic options for early tumor stages. For multifocal and metastatic diseases, systemic therapy is recommended. While Sorafenib had been the standalone HCC first-line therapy for decades, recent developments had led to the approval of new treatment options as first-line as well as second-line treatment. Anti-PD-L1 directed combination therapies either with anti-VEGF directed agents or with anti-CTLA-4 active substances have been implemented as the new treatment standard in the first-line setting. However, data from clinical trials indicate different responses on specific therapeutic regimens depending on the underlying pathogenesis of hepatocellular cancer. Therefore, histopathological examinations have been re-emphasized by current international clinical guidelines in addition to the standardized radiological diagnosis using contrast-enhanced cross-sectional imaging. In this review, we emphasize the current knowledge on molecular pathogenesis of hepatocellular carcinoma. On this occasion, the treatment sequences for early and advanced tumor stages according to the recently updated Barcelona Clinic Liver Cancer classification system and the current algorithm of systemic therapy (first-, second-, and third-line treatment) are summarized. Furthermore, we discuss novel precautional and pre-therapeutic approaches including therapeutic vaccination, adoptive cell transfer, locoregional therapy enhancement, and non-coding RNA-based therapy as promising treatment options. These novel treatments may prolong overall survival rates in regard with quality of life and liver function as mainstay of HCC therapy.

## 1. Introduction

Liver cancer is worldwide one of the major health problems and recommends high standards in diagnostics and medical care. In 2020, liver cancer was the sixth most common cancer, with 905,677 new cases per year [1,2,3,4,5]. Despite advances in diagnosis and treatment, mortality from liver cancer remains high. Liver cancer is the second most lethal tumor, following pancreatic cancer, with 830,180 deaths yearly. The relative five-year survival rate is only 18% [6]. Within primary liver tumors, hepatocellular carcinoma (HCC) represents the most critical tumor entity. It accounts for approximately 75% to 90% of all cases [1,2,3,4,5]. After HCC, intrahepatic cholangiocarcinoma is the second most frequent entity, with about 10%-15% of all cases [1]. Besides, there are rare types of primary liver cancer, including fibrolamellar carcinoma, angiosarcoma, and hepatoblastoma.

Regardless of the etiology (hepatitis C virus (HCV) infection, hepatitis B virus (HBV) infection, metabolic syndrome, chronic alcohol abuse, hemochromatosis, α1-antitrypsin deficiency), liver cirrhosis is the leading risk factor for developing HCC. Approximately one-third of all patients with liver cirrhosis will develop HCC during their lifetime. In patients with HBV-associated liver cirrhosis, approximately 2% per year, and in patients with HCV-associated liver cirrhosis, 3–8% develop HCC [7,8,9,10]. Besides viral hepatitis, non-alcoholic fatty liver disease (NAFLD) and non-alcoholic steatohepatitis (NASH), have become significant risk factors for HCC in recent years, but with high heterogeneity of newly diagnosed HCC annually [11,12]. 

Depending on tumor stage and liver function, a wide range of treatment options for HCC is indicated. The prognosis of patients with HCC depends highly on performance status and liver function. Therefore, HCC in liver cirrhosis is classified by the updated Barcelona-Clinic-Liver cancer (BCLC) Classification system. As already mentioned, the treatment concepts suggested by the BCLC classification system strongly depend on the preserved liver function in these patients as the underlying condition is liver cirrhosis.

The treatment concept is curative in the very early stage (BCLC stage 0) and early stage (BCLC stage A). Curative therapeutic approaches are liver resection, local ablation therapy, or transplantation (Figure 1). The 2022 updated BCLC classification system addresses some additional issues in comparison to previous versions. For BCLC B stage, the 2022 BCLC version stratifies 3 groups of patients according to tumor burden and liver function. This subgroup includes patients with well-defined HCC nodules that are still candidates for liver transplantation if they meet the “extended criteria for liver transplantation”. Downstaging has emerged as a reliable tool to select patients for liver transplantation. The goal is reduction of tumor burden with the residual viable tumor reaching the Milan criteria as the most common endpoint of downstaging. Besides reduction of viable residual tumor biological response in terms of decrease of AFP values (decrease at least <500 ng/mL) have been sought to expand Milan criteria.

The second subgroup comprises patients that are clear candidates for TACE without the option of liver transplantation. They present with preserved portal vein flow, defined tumor burden and feasible access to the feeding tumor arteries. 

If patients neither meet the “extended criteria for liver transplantation” nor criteria suggesting therapeutic success of TACE systemic therapy should be considered. In this third subgroup of patients in BCLC-B, diffuse infiltration with extensive tumor burden TACE is not beneficial and systemic therapy should be recommended.

Another step forward in the updated BCLC classification system is the incorporation of an expert clinical decision-making component and stratification after radiological progression after initial diagnosis. Radiological progression and/or treatment-related adverse events may lead to treatment recommendations even if BCLC stage has not changed (see Figure 1, BCLCp-B; BCLCp-C1; BCLCp-C2). Therefore, the term “treatment stage migration” has been introduced. 

In case of multifocal tumor manifestation, macrovascular portal vein invasion, extrahepatic tumor spread (BCLC stage C), or slightly reduced performance status (ECOG 1-2), the guidelines recommend systemic therapy. In the first line setting of systemic therapy, physicians choose either therapies with tyrosine kinase inhibitors (TKI) such as sorafenib and lenvatinib, or immune-oncological (IO) approaches with atezolizumab and bevacizumab or durvalumab/tremelimumab [10,15,16] (Figure 2). In the pivotal trials, subgroup analysis demonstrated patients with HCC secondary to viral hepatitis as the primary beneficiaries of the initiation of IO-based therapies [16]. On the other hand, patients with HCC based on NAFLD or NASH appeared with higher beneficials from therapy with TKI, such as lenvatinib or sorafenib.

In this regard, it is evident to identify clinical and molecular subtypes for a better tailored and individualized medicine in systemic therapy of HCC. Due to the global disease burden, it is of utmost importance to better understand HCC pathogenesis to develop new treatments and approaches with promising therapeutic targets. This review aims to provide an overview of the current state of molecular pathogenesis and treatment options for HCC.

## 2. Etiology and Risk Factors in HCC Pathophysiology 

As already introduced, HCC is an ever-growing health problem and the second leading cause of cancer-related death worldwide. It is the ninth most common cancer in women and the fifth most common cancer entity in men [17]. Various risk factors are known for the development of HCC (Figure 3). Chronic hepatitis B or C virus infections, hereditary diseases, e.g., hemochromatosis, α1-antitrypsin deficiency as well as tyrosinemia or glycogen storage disease type 1a play a role in HCC pathogenesis [5,18]. In addition, lifestyle components, such as excessive alcohol consumption or nicotine abuse, favor the development of this neoplasia [18,19,20,21]. 

The common feature of all these factors is induction of liver cirrhosis as a possible complication [5]. Clinical studies have shown that about 8% of all patients with liver cirrhosis develop HCC [22]. Hence, patients with liver cirrhosis are defined as high-risk group to be included in structured surveillance for early detection of HCC. The Guidelines published by the American and European Association for the Study of liver disease (AASLD and EASL) recommend in adults with cirrhosis a close surveillance with abdominal ultrasonography at intervals of 6 months and an optional alpha-fetoprotein (AFP) screening [23].

### 2.1. Viral Risk Factors

Approximately 54% of HCC cases are due to HBV infection [22]. Vaccination-programs reduce the transmission risk by 90%. As a result, the incidence of HCC, especially in male adolescents, has been significantly dropped [24,25]. Despite these vaccination initiatives against HBV, the risk is still very high in endemic areas, such as Africa or East Asia. In these areas, HCC pathogenesis is initiated in average 10 years earlier than in patients in Europe and North America. This is most likely due to early infection, mostly through vertical transmission during pregnancy or birth. In up to 90% of cases, chronic viral disease develops. In contrast, when HBV prevalence is low, as in most Western industrialized nations, horizontal transmission of the virus predominates. 

Another 31% of HCC are due to HCV infection. In contrast to HBV, HCV is usually not acquired at birth but later. Therefore, HCV-related liver tumors usually only occur at later ages [22]. Transmission often occurs predominantly through contaminated application equipment in drug abuse or through infected blood products. The degree of HCV-induced liver damage determines the risk of HCC. Without confirmed liver cirrhosis, HCV-infected patients show only a slightly increased for HCC with an annual incidence of 0.48% [5]. In contrast, patients with liver cirrhosis based on chronic HCV infection have a high risk to develop HCC with an annual incidence of 3–8% [7,9]. The risk of HCC in patients with chronic HCV decreases significantly to 0.33%-0.9%/year after successful antiviral therapy with sustained virological response (SVR) [11,26]. With already existing cirrhosis of the liver, the risk of HCC continues to be high despite SVR. The literature indicates an HCC risk of 1.5% to 2.4%/year for these patients [9,11,27]. 

### 2.2. Aflatoxin

In addition to viral factors, toxins also play a crucial role in the development of HCC. Widespread contamination of cereals, maize, and tree nuts with aflatoxins is known, especially in East Asia and southern African countries, particularly in rural areas [28]. Four different chemical groups are distinguished, aflatoxin B1, B2, G1 and G2, of which B1 is considered the most dangerous. Aflatoxins induce mutations of the tumor suppressor p53 at the third base of codon 249, which are detectable in 30 to 60% of HCC patients exposed to aflatoxin. There is a 4-fold increased risk of cellular degeneration upon mutation of p53. In addition, in most areas with high aflatoxin exposure, there is also a high prevalence of HBV. In combination with chronic HBV infection, Aflatoxin can lead to up to a 60-fold increased probability of developing HCC compared to healthy individuals not exposed to aflatoxin. Thus, a synergistic effect of the two risk factors can be assumed [28].

### 2.3. Alcohol Consumption

Inflammatory processes, hepatocyte necrosis and regenerative processes associated with oxidative stress condition the development of hepatic cirrhosis with regular alcohol consumption [18]. In most cases, HCCs in viral hematogenesis are also cofounded by alcohol consumption. This also highlights the synergistic effect of alcohol abuse and HBV or HCV disease [16,26]. In a prospective study in patients with chronic HCV infection and liver cirrhosis, even with simultaneous mild to moderate alcohol consumption, there is a significantly increased HCC risk with a cumulative 5-year HCC incidence of 23.8%. In comparison, the incidence of HCC in patients who did not consume alcohol at all was 10.6%, only [29]. However, even without the pre-existence of infections, alcohol is a major risk factor for HCC pathogenesis. On the one hand, synergistic effects between alcohol and obesity are assumed. On the other hand, ethanol increases HCC risk in relation to concentration, especially in societies with an otherwise low risk of HCC and a low prevalence of viral hepatitis [28].

### 2.4. Metabolic Syndrome, Non-Alcoholic Fatty Liver Disease (NAFLD) and Non-Alcoholic Steatohepatitis (NASH)

In Western world countries, metabolic diseases are risk factors for hepatocarcinogenesis as well [30]. Metabolic diseases favor liver steatosis as a precursor to fibrosis or cirrhosis. The extent to which steatosis or fibrosis can be considered independent risk factors for the development of HCC has not been determined, conclusively [28]. In addition, NAFLD is considered a risk factor for HCC, as well [18]. Especially in industrialized nations, the proportion of patients with an HCC based on the presence of NAFLD without cirrhosis is increasing [30]. The most severe form of NAFLD, the NASH based cirrhosis, may further increase the HCC risk [18]. NASH is characterized by steatosis, inflammation, degenerative changes, and fibrosis. Progression to marked fibrosis occurs in 15 to 50% of cases, progression to cirrhosis in 7 to 25% [28]. The incidence of HCC is reported to be 0.02% per year in patients with NASH but without cirrhosis. The HCC increases up to 1.5% per year in the presence of cirrhosis. Therefore, it is recommended that patients with NASH without cirrhosis, i.e., with advanced liver fibrosis, should be monitored closely [31,32]. In the presence of NAFLD or NASH without evidence of advanced fibrosis, the risk of HCC is lower but still present. Recommendations for surveillance in this population are missing [10,33,34]. 

Other, rare diseases may also be at increased risk for developing HCC. Among others, the presence of glycogen storage disease, in particular type Ia and Ib, is associated with the occurrence of hepatic adenomas [35]. In addition, patients with M. Gaucher show a significantly increased risk of developing HCC even without cirrhosis and tyrosinemia type I (HT I), when left untreated, is the most common cause of HCC in children [36,37].

## 3. Somatic Mutations in HCC Pathogenesis 

Somatic mutations are the most common cause for a healthy cell turning into an abnormal cell. Single gene mutations are most commonly the initiators (driver genes) of consecutive accumulations that eventually lead to cancerogenic progression. Apart from Melanoma and lung cancer, which have the highest proportion of mutations per genome, HCC has an intermediate number of mutations per genome [38]. However, among other solid tumors, the development of HCC is most strongly correlated with influential environmental factors (e.g., virus-related) [39]. HBV and HCV are viruses either integrating their genetic material into the host genome or inducing double-strand breaks resulting in numerous genetic alterations. Molecular studies have identified the most frequent alterations in HCC, including mutations in the TERT promoter, TP53, CTNNB1, and epigenetic aberrations [40,41]. In the following, the commonly observed somatic mutations in HCC are discussed in detail (see Table 1). 

### 3.1. The Tumor Suppressor p53

*TP53* is the most frequently mutated gene in human cancer [42]. In its physiological state, p53 executes a monitoring and tumor-suppressive function [43,44,45]. Due to the accurately regulated expression, degradation, and fined tuned protein isoform ratio of p53, the cell can rapidly react to DNA damages and initiate an appropriate cellular response [46]. Its versatility allows p53 to operate as a transcription factor, a recruiting factor for DNA-repair proteins at damaged sites, or a direct protein interaction partner [47,48,49]. Thus, p53 also executes many regulatory functions [50]. Most notably, p53 is known for its central role in triggering various types of cell death. Among them, intrinsic apoptosis is initiated by direct interaction of p53 with anti-apoptotic Bcl-2-family members, namely Bcl-2 and Bcl-XL. In response to inhibition of Bcl-2 and Bcl-XL, pro-apoptotic Bax and Bak form pores in the mitochondrial outer membrane (MOM), which leads to cytochrome C release and formation of the cytosolic death platform, the apoptosome [49]. p53 activation is also known to promote upregulation of genes involved in accumulation of intracellular iron and reactive oxygen species (ROS). Abnormally high levels of Fe^2+^ lead to generation of hydroxide anions and thus lipid-peroxidation. The latter proves to be highly toxic for cells. This type of programmed cell death is known as ferroptosis [51]. Especially the inactivation of glutathione peroxidase 4 (*GPX4*) is especially a major driver of ferroptosis. Normally, GPX4 catalyzes incurring lipid-peroxidases into harmless lipid-alcohols by using glutathione as cofactor. P53 is capable of either boosting glutathione production via upregulation of p21, or to prevent further glutathione supply by suppressing SLC7A11. Particularly, mutated p53 preferably binds to NRF2 protein and massively reduces cellular glutathione concentration, thus indirectly decreasing GPX4 activity [52].

Other p53 family members including p63 and p73 also possess essential regulatory properties. Either in concert with or as a substitution for p53, p63 and p73 tightly regulate proliferation, metabolic activity, cell differentiation and initiation of apoptosis and/or ferroptosis [51,53,54,55,56]. However, p63 and p73 have a lower mutational rate than p53 [57]. 

**Table 1 biomedicines-10-03202-t001:** Pathophysiological factors involved in HCC carcinogenesis (gene mutation, miRNA, lncRNAs, epigenetic changes, alteration in signaling pathways, etc.). The right column contains the corresponding references.

Category	Gene/Pathway	Incidence	Reference
Somatic mutations	Tumor suppressor p53	AFB1 induced R249S	[58,59,60]
		AFB1 induced V157F	[58,60,61]
		HBV promoted Mdm2 promoter polymorphism (SNP309, rs2279744)	[61,62]
		*ZNF498*	[51]
		*SLC7A11*	[52]
		*GPX4*	[52]
		*ATM*	[63]
		*RPS6KA3*	[63]
	Telomerase promoter	*TERT* promoter C to T transition (−228 to −250)	[64,65]
	HBV-promoted	HBV-DNA integration in *TERT*	[66,67,68]
		*MLL4*, *KMT2B*	[69,70]
		*CCNE1*, *CCNA2*	[69,70]
		*SENP5*	[71]
		*RARβ*	[72]
		*ROCK1*, *ARHGEF12*	[69,70]
		*FN1*	[73]
		*CYP2C8*	[73]
		*PHACTR4*	[73]
		*SMAD5*	[73]
	Others	*CTNNB1*, *AXIN1*	[74]
		*mTORC1*, *ARID1A*, *ARID2*	[74,75]
		*TTN*	[74,75]
		*JAK1*	[74,75]
		*KEAP1*	[74,75]
		*NRF2*, *NFE2L*2	[76,77]
		*KRAS*, *NRAS*, *BRAF*	[78]
		*EGFR*	[78]
		*IDH1*, *IDH2*	[78]
Epigenetic changes	DNA methylation	genes involved in Ras/Raf/ERK and Wnt/β-catenin pathway	[79]
		*GSTP1*	[80]
		*FANCB*	[81]
		*KIF15*	[81]
		*KIF4A*	[81]
		*ERCC6L*	[81]
		*UBE2C*	[81]
		*CDKN2A*	[82]
		*p16*	[83]
		*DNMT1*, *DNMT3A*, *DNMT3A2*	[84]
	Histon and chromatin modification	H3K3me3, H3K27ac, H3K9ac, H3K4me2	[85]
		H3Kme27, H3K4me3	[85,86]
		*Ash2*	[87]
		*LSD1*	[87]
		*MMP1/MMP3*	[88]
		*VASH2*	[87,88]
		*EZH2*	[89]
		*SMARCD1*	[90]
		*ARID2*	[91]
		Hbx-antigen interaction with CREB-binding protein/p300	[92]
	Micro RNAs	miR-1/-122/-124/-132/-148/-200/-205	[93,94]
		miR-210-3p	[95]
		miR224	[96]
		miR-196a and miR-196b	[97]
		miRNA-1468	[96,97]
		miRNA144	[98]
		miR-342-3p	[99]
		miR-1/-124/-214/-34-A/-449	[100]
	Long-non-coding RNAs	LINC01234	[101]
		PTTG3P	[102]
		HULC	[103]
		HEIH	[104]
		MVIH	[103,104,105,106,107]
		MAIT	[102]
		MIR31HG	[108]
Pathway dysregulations	Wnt/β-catenin	*CTNNB1*	[78]
		*AXIN1*	[109,110]
		AXIN2	[109,110]
		APC	[109,110]
		*UBE2T*	[111]
		*TFAP4*	[112]
		*DDX39*	[113]
		circ_0004018	[114]
		circ_0003418	[115]
	Receptor tyrosine kinase	mTORC1, mTORC2	[116]
		*PTEN*	[117,118]
		*LZTS2*	[119]
		*HJURP*	[120]
		*RSK2*	[121]
		miR155-5p/-494/-493/-519a	[122]
	VEGF	*HIF-1*, *HIF-2*	[123]
		*ERO1α*	[124]
		lncRNA PAARH	[125]
	JAK/STAT	IL-6	[126]
		GP130	[127]
		JAK1	[128]
		IL6R	[128]
		IL6ST	[128]
	TGF-β	SMAD2/3	[129,130]
		*PTPRε*	[129]
		exosomes	[131]

Due to the numerous tasks that p53 takes over, a disordered protein isoform proportion [132], a subfunction, or complete breakdown can only be compensated by the cell with great difficulty or not at all [53]. In HCC, the frequency of p53 gene mutation ranges from 15% to 40%, depending on the underlying etiology [133]. For this reason, *TP53* mutation-related HCC is highly dependent on geography and ethnicity [2,134,135]. Dietary exposure to fungal aflatoxin B1 (AFB1), for instance, induces a multitude of genetic alterations in the liver [58]. Since AFB1 metabolizes into a reactive epoxide species named AFB1-E, it can rapidly react to guanine residues, leading to DNA damage and mutations [60]. The most-reported aflatoxin B1 induced mutation in *TP53* locates at codon 249, evoking a missense mutation from arginine to serine (R249S) [59]. This codon lies within the DNA-binding domain. Alteration of arginine 249 into any other amino acid causes a reduced binding affinity to the corresponding p53-response elements in the DNA. Another AFB1-induced hotspot mutation in *TP53* is V157F, which provokes a poor prognosis for patients with HCC. Together with HBV chronic inflammation, those mutations may advance the development of HCC [61]. Conversely, previous chronic HBV infection was shown to promote the transition from G/C to A/T after additional AFB1 exposure [60]. In addition to mutations that directly affect the integrity of p53, genes involved in the negative feedback loop also contribute to disease development. In principle, the 309T > G polymorphism (SNP 309, rs2279744), located in the intronic p53-responsive promoter of *MDM2*, has the effect of increasing Mdm2 protein levels and is associated with the early onset of HCC in patients with chronic HCV infection [62,136]. Recently, KRAB-type zinc-finger protein (ZNF498) was found to be upregulated and therefore suppress apoptosis and ferroptosis by attenuating p53 Ser46 phosphorylation in hepatocellular carcinogenesis [51].

Other genes belonging to the p53 pathway and encoding p53-activating kinases, namely *ATM* and *RPS6KA3*, were also recurrently mutated [63]. 

### 3.2. The Telomerase Promoter

In each cell of the human organism, the genetic material is subdivided into 46 chromosomes. So-called telomeres cover the ends of chromosomes to protect genomic integrity. These telomeres consist of several kilobase pairs of long repetitive sequences, that shorten by approximately 100 base pairs with every cell division. The telomerase complex controls the telomere length. The telomerase complex comprises the telomerase reverse transcriptase (*TERT*), telomerase RNA component (*TERC*), and several proteins such as the shelterin components *TRF1*, *TRF2*, *TIN2*, *RAP1*, *TPP1*, and *POT1* [137]. The liver is a quiescent organ and therefore, shows low physiological telomerase expression [64]. During chronic liver injury and inflammation, hepatocytes undergo progressive telomere shortening. However, in the absence of telomerase activity, chromosome erosion and genomic instability are intercepted by p53-induced replicative senescence and apoptosis in most critical cases [138,139,140]. Thus, tumor progression is usually impaired in healthy hepatocytes [141]. Reactivation of the telomerase and the associated telomere regeneration gives hepatocytes the ability to overcome telomere-shortening-induced senescence and proliferate continuously [142,143]. *TERT* and *TERC* upregulation are critical events in liver tumor progression, observed in 90% of the HCC patients [64]. The most prevalent somatic mutation resulting in elevated telomerase expression is a C to T transmission in the *TERT promoter* at positions -228 and -250 upstream of the start codon [65,66]. Several studies indicate a preferential HBV-DNA integration into *TERT* or *TERT*-near regions [67,68,144]. While *PROX1* (positive modulator of *TERT*) is downregulated by the presence of Hepatitis B virus X (HBx) presence [84], other modulators of *TERT* expression such as *NCOA3* are enhanced in their specific binding to the *TERT* promoter at the -234 to -144 region [145]. Further, the methyltransferase DNMT3B is activated upon *TERT* upregulation, resulting in a remodeling of the DNA-methylation landscape. For example, HCC-specific DNA methylation remodeling by DNMT3B leads to AKT activation within the PI3K/AKT pathway, resulting in inhibition of Bax and ultimately suppression of mitochondria-mediated apoptosis [146]. Therefore, *TERT promoter* mutations represent one of the most critical driver mutations in pathologic liver progression.

### 3.3. Hepatitis B Virus Integration Mediated Mutations

The ~3.2 kilobase (kb) pairs long genome of HBV consists of circular double-stranded DNA and comprises four overlapping open reading frames (ORF), namely *preS/S*, *P*, *preC/C*, and *X*, that result in seven different gene products [147,148]. These viral proteins are mainly responsible for integrating viral DNA into preferential sites of the human genome. HBV integration can cause several genomic alteration events such as (1) HBV-promoter induced transcription of host genes, (2) viral-host transcript fusion, which may lead to activation of proto-oncogenes or inactivation of tumor suppressor genes, and (3) gene disruption [61]. The latter mainly occur in exons or regulatory regions, including *TERT* (Telomerase reverse transcriptase), *MLL4* and *KMT2B* (Methyltransferases, chromatin remodeling), *CCNE1* and *CCNA2*, (Cyclin E1 and A2, cell cycle control), *SENP5* (Sentrin-specific protease (5) [71], *RARβ* (Retinoic acid receptor beta), *ROCK1* and *ARHGEF12* (Rho-Rock associated pathway) [63,67,69,70,72,149]. Additional recurrent integration sites are found in *FN1* (Fibronectin 1), *CYP2C8* (cytochrome P450, family 2, subfamily C, polypeptide (8), *PHACTR4* (Phosphatase and actin regulator (4), and *SMAD5* (SMAD family member (5) [73]. Therefore, HBV virus integration predisposes cancer initiation by promoting deletions or translocations of the host genome and increasing chromosomal instability [150].

### 3.4. Other Somatic Mutations

Other mutations consistently found in HCC patients include *CTNNB1* and *AXIN1*, which are involved in the Wnt/β-catenin signaling pathway [74]. Additional mutations are found in *mTORC1*, *ARID1A*, and *ARID2* (chromatin remodeling) [75,76], in *TTN* (chromosome segregation), in *KEAP1* (involved in ubiquitination), in *JAK1* (JAK/STAT signaling pathway). Also, mutated *NRF2* and *NFE2L*2 are upregulated upon oxidative stress [77,151]. Genes that are rarely mutated but rather typical in HCC include *PIK3CA* and *PTEN* (PI3K/AKT signaling pathway), *KRAS*, *NRAS*, and *BRAF* (RAS/MAPK signaling pathway), *EGFR* (growth factor signaling pathway), IDH1 and IDH2 (NADPH metabolism) [78].

## 4. Epigenetic Regulation in HCC 

Epigenetics describes the heritable changes in phenotype provoked by alterations in gene expression without affecting the DNA sequence. Epigenetic dysregulation includes changes in transcriptional regulation, chromosomal stability, and altered differentiation or proliferation status. Epigenetic alterations can strongly contribute to tumorigenesis in numerous cancer types, including HCC. Epigenetic mechanisms comprise a) changes in methylation, hydroxy methylation, and acetylation of global or specific DNA regions, b) modifications of histone proteins that specify the packaging of the DNA and c) gene expression regulation through non-coding RNA molecules [152] (see also Table 1).

### 4.1. DNA Methylation

The most profound way cells regulate their gene expression is through methylation changes on individual CpGs or whole CpG islands in the DNA. On the one hand, hypermethylation of promoters or promoter-near regions downregulates the expression of specific genes. On the other hand, hypomethylation correlates with increased gene activity. Alterations in DNA methylation are frequently observed in HCC and play an essential role in carcinogenesis [153]. Most hypermethylated regions are detected in promoters of tumor suppressor genes, whereas hypomethylation occurs in regions responsible for cell-cycle and proliferation regulation (oncogenes). 

Epigenetically altered genes often correlate with certain etiological factors, such as HBV and HCV infection. Deng et al. have shown 15 genes to be significantly downregulated upon HCV infection. These genes are mainly involved in the Ras/Raf/ERK and Wnt/β-catenin pathway and strongly contribute to the progression of HCC [79].

Moreover, HBV may enhance or maintain *GSTP1* and *E-cadherin* promoter methylation, affecting hepatocarcinogenesis [80]. HBV and HCV independently, 115 genes were significantly up-or down-regulated in HCC tissue, also associated with poor prognosis. In particular, the upregulated genes are involved in cell division, cell cycle, and proliferation. The most considerable genes hypomethylated on CpGs around their promoter are *FANCB*, *KIF15*, *KIF4A*, *ERCC6L*, and *UBE2C* [81]. *CDKN2A*, a key regulator of G1 cell cycle arrest, is hypermethylated and inactivated in many cancer types [82]. It has also been reported that *CDKN2A* inactivation influences the epigenetics of p16. p16, in turn, is a profound tumor suppressor and is inhibited in HCC [83]. HBV infection and HBx antigen expression further showed a total increase in DNMTs activity. Methyltransferases such as *DNMT1*, *DNMT3A*, and *DNMT3A2* are significantly upregulated, contributing to further epigenetic changes [84]. 

This chapter focuses on the importance of the methylation-related changes and their role in tumor progression. Although epigenetic studies in cancer are still a comparatively novel area of research, the massive data generation through Next Generation Sequencing emphasizes the potential of essential strategies in discovering new biomarkers and novel therapeutic approaches [85,154,155]. 

### 4.2. Histon Modification and Chromatin Remodeling

Histones are the packaging material of the DNA that can occupy different conformations. Histone-DNA complexes (nucleosomes) are either densely (heterochromatin) or less densely packed (euchromatin). Consequently, it leads to either weak or strong gene activation. In several cancer models, including HCC, histone H3 modifications were investigated intensively. Major post-translational modifications comprise histone methylation, acetylation, phosphorylation, and ubiquitination. Histone methylation and acetylation changes were mainly investigated [156]. Liu et al. reported four histone modifications that frequently occur in HCC. These are H3K3me3, H3K27ac, H3K9ac, and H3K4me2 [85]. Highly methylated H3Kme27 and H3K4me3 have been shown to correlate with poor prognosis and aggressive behavior of HCC [86,157,158]. Increased histone methylations may be explained by the markedly increased activity of the Ash2 methylation complex. Demethylating enzymes such as LSD1 are predominantly absent in HCC [87]. 

Significant characteristics of tumor aggressiveness include cell invasion and metastasis. Mixed lineage leukemia (*MLL*) is a methyltransferase responsible for the trimethylation of H3K4. H3K4me3, in turn, is necessary for the interacting-complex MLL-ETS2 to occupy the promoter of *MMP1/MMP3* and ultimately increase their expression [88]. Therefore, activating matrix metalloproteases by the HGF signaling pathway induces metastasis [156]. It has further been reported that putative vasohibin 2 (*VASH2*) abnormally overexpresses in HCC to promote cell proliferation and inhibit apoptosis [159,160]. 

Another important player for chromatin remodeling is EZH2. It functions as a methyltransferase and belongs to the Polycomb Repressive Complex 2 (*PRC2*). EZH2 is strongly upregulated in pathogenic liver tissue and fundamentally alters the chromatin structure. It promotes HCC progression by silencing β-catenin antagonists, thus activating the Wnt/β-catenin signaling pathway. The Switch/Sucrose Non-Fermentable complex (*SWI/SNF*) with a similar chromatin remodeling function is also frequently linked with liver tumorigenesis. According to the current state of research, it is still inconclusive whether *SWI/SNF* acts as a tumor suppressor or oncogene [89]. The SWI/SNF complex comprises various active subunits (*ARID1A*, *ARID1B*, *SMARCA2*, *SMARCA4*, *SMARCD1*). Mutation or dysregulation of these active subunits affects the cell physiology differentially. The subunit *SMARCD1* is considered an oncogene since its excess in the cell promotes liver cancer growth by activating the mTOR signaling pathway [90]. In contrast, ARID2, which encodes a regulatory subunit, mutates 18.5% of HCV-related liver cancer and is believed to act as a tumor suppressor [91]. In addition, HBV infection also affects histone modification. The incorporation and direct interaction of Hbx antigen with CREB-binding protein/p300 increase the acetyl-transferase activity, which alters gene transcription and promotes tumorigenesis [92].

### 4.3. Micro RNAs

Micro RNAs (miRNAs) are small non-coding RNA molecules with an average length of 22 ribonucleotides that regulate the cell’s translational output. In complexation with the RNA-induced silencing complex (RISC), miRNA and RISC can bind complementary mRNA sequences. In complexation, they degrade and downregulate specific gene activities post-transcriptionally [93]. Since miRNAs are simply detectable in sera of patients, many studies drew attention to the altered levels of specific miRNAs in different cancer types, including HCC. MiRNAs themselves belong to epigenetic gene-regulators. However, the transcriptional activity of the corresponding miRNA genes, in turn, can be regulated by epigenetic changes directly on the DNA (e.g., methylation, acetylation) [94]. An example of miRNA genes affected by downregulation due to HBV infection includes miR-1/-122/-124/-132/-148/-200 and miR-205 [161,162]. Their dysregulation has significant effects on a variety of different signaling cascades that often modulate cell proliferation, differentiation, migration, and maintenance. Thus, high levels of miR-210-3p were detected in sera of HCC patients correlating with increased micro-vessel density in HCC tissue.

Previously harvested exosomes from HepG2 cell culture containing miR-210-3p promoted in vitro tubulogenesis of endothelial cells. Lin et al. further revealed the molecular pathomechanism: miR-210 represses *SMAD4*, an essential transcription factor of the TGF-β signaling pathway. SMAD4 repression enhances angiogenesis by increasing the proliferation of endothelial cells [95]. miR-224 are at high expression levels in HCC patients with poor prognoses. miR224 targets glycine methyltransferases. Their downregulation consequently leads to globally altered DNA methylation and increased proliferation in liver cells [96]. Another signaling cascade influenced by the deterioration in liver cells is the JAK/STAT pathway through abnormally high miR-196a and miR-196b levels. These miRNAs arouse the progression of HCC by targeting *SOCS2* and thereby stimulate JAK/STAT signaling [97]. The clinical analysis further revealed that downregulation of *CITED2* and *UPF1* stimulates the peroxisome proliferator-activated receptor-γ (PPAR-γ)/Akt inducing pathway, which inhibits apoptosis by repressing pro-apoptotic Bcl-2 proteins while stimulating cell growth and proliferation through mTORC1 and mTORC2 [163,164]. 

In contrast, some miRNAs suppress typical tumor characteristics. For example, overexpression of miRNA144 suppresses cell proliferation, migration, and invasion by inhibiting CCNB1 [98]. Therefore, miR144 is believed to be a tumor suppressor that is downregulated during tumor progression. Likewise, tumor suppressor miR-342-3p is high in HCC and low in regressing tumors [99]. Further tumor-suppressive miRNAs, including miR-1, miR-124, miR-214, miR-34-A, and miR-449, target downstream mRNA involved in tumor progression and usually are downregulated in HCC [100]. Both etiologic factors and epi-/genetic alterations can completely change the miRNA profile of a cell. Since circulating miRNAs are easily accessible through liquid biopsies, they are promising biomarkers for detecting early-stage hepatocellular carcinoma [165,166]. 

### 4.4. Long Non-Coding RNAs

Long non-coding RNAs (lncRNA) are another class of RNA molecules that regulate gene expression post-transcriptionally. Most of these lncRNAs are transcribed at a low level and, therefore, hardly detectable. However, a change in lncRNA-signature, measured via blood, can indicate a tumorigenic progression in the liver [167]. As a result, numerous studies have shown that lncRNAs play an essential role in the development and progression of cancer, including HCC [168,169,170]. LncRNAs regulate gene expression and protein synthesis by diverse mechanisms. From a physiological state, divergent expression of lncRNA can promote oncogenes and repress tumor suppressor genes involved in hepatocarcinogenesis, proliferation, cell growth, invasion, and metastasis [171].

The Cancer Genome Atlas (TCGA) analysis shows that upregulation of lncRNA LINC01234 is associated with poor prognosis in patients with HCC. LINC01234, in concert with melanoma-associated antigen A3 (*MAGEA*), was demonstrated to bind to tumor suppressor miR-31-5p and eventually mediate proliferation, invasion, and cis-platin chemoresistance in HCC cell lines [101]. Huang et al. have found another lncRNA, named PTTG3P, to promote cell growth and metastasis by upregulating *PTTG1* and activating PI3K/AKT signaling [102]. LncRNA HULC was first characterized in 2007 as Highly Upregulated in Liver Cancer [103] and is elevated in HBV-related-HCC. Thereby, virus antigen HBx protein facilitates cell proliferation via downregulation of *CDK4*- and *CDK6*-inhibitor p18 [105,106]. Further, lncRNAs significantly upregulated in HCC are lncRNA-HEIH, -MVIH, and -MAIT. High Expression in Hepatocellular carcinoma (lnc-HEIH) promotes tumor growth by directly associating with enhancer of zeste homologue 2 (*EZH2*) to suppress its enzymatic activity [104]. LncRNA-associated microvascular invasion in HCC (lnc-MVIH) activates angiogenesis by promoting tumor growth and intrahepatic metastasis. It inhibits the expression of phosphoglycerate kinase 1 (*PGK1*). PGK1, in turn, is an essential enzyme in the aerobic glycolysis pathway and is known to suppress angiogenesis [104,107,172]. Massive ATP consumptions due to high proliferative rates urge hepatocellular carcinoma cells to shift towards aerobic glycolysis. Even in the presence of sufficient oxygen, liver carcinoma cells convert glucose into pyruvate and eventually to lactate rather than import pyruvate into the citric acid cycle [173,174,175]. Among others, this phenomenon was described in the early 1920s by Otto Warburg and his co-workers. A process that is observed in a multitude of different cancer types [176]. 

Some lncRNAs can regulate gene expression indirectly via miRNA-silencing. MAIT promotes the proliferation and invasion of HCC cells by sponging miR-214, thus being inaccessible to the RISC complex [102]. Another lncRNA operating on the same principle but rather tumor-suppressive is lncRNA MIR31HG. It sponges oncogenic miR-575, thereby preventing inhibition of Suppression of Tumorigenicity 7-Like (*ST7L*) and inducing tumor repression. In contrast to healthy tissue, MIR31HG is significantly downregulated in HCC [108]. Low lncRNA MIR22HG levels (also registered as a tumor suppressor) were correlated with a malignant HCC phenotype [177]. 

Like miRNAs, deviations in lncRNA signatures are easily detectable and can serve as potential biomarkers to identify malignant traits in HCC. 

## 5. Dysregulation of Key Signaling Pathways 

Signaling pathways count for the most complex and vulnerable systems that preserve the inter-and intracellular communication. Extracellular stimuli received at the cell surface are translated into intracellular responses. Signaling pathways regulate the physiology of a cell in terms of proliferation, cell growth, differentiation, and apoptosis. Somatic mutations, chromosomal aberrations, and epigenetic alterations are mainly responsible for the constitutive activation of certain signal cascades [178]. Such dysregulations mostly trigger typical tumorigenic characteristics, also known as hallmarks of cancer [179]. There are predominant pathways recurrently disordered in HCC. They include growth signaling pathways such as insulin-like growth factor (IGF), epidermal growth factor (EGF), platelet-derived growth factor (PDGF), fibroblast growth factor (FGF), and hepatocyte growth factor (HGF/c-MET). Key pathways in cell differentiation include Wnt/β-catenin, JAK/STAT, Hippo, Hedgehog, and Notch. Tyrosine-kinase-dependent signaling pathways include Ras/Raf/MEK/ERK and PI3K/AKT/mTOR [180,181]. The most altered signaling pathways involved in liver pathogenesis are discussed below (Figure 4).

### 5.1. Wnt/β-Catenin Signaling Pathway

The Wnt/β-catenin pathway is one of the best-studied signaling cascades. In terms of liver homeostasis and regeneration in adults, it can regulate cell differentiation, proliferation, and initiate repair mechanisms [182,183]. The Wnt/β-catenin pathway is also referred to as the “canonical Wnt pathway”. Signal transduction is initiated by Wnt-ligand binding to a large membrane-anchored complex consisting of LRP-5 or LRP6 and one member of ten different *frizzled* receptor proteins [184]. In the absence of Wnt ligand, β-catenin complexes with negative regulators such as adenomatous polyposis coli (APC) and AXIN proteins, which mediate N-terminal phosphorylation of β-catenin. The subsequent ubiquitination by β-TRCP leads to its proteasomal degradation. Stable binding of the Wnt ligand to the *frizzled* receptors phosphorylates Dvl. Then, Dvl recruits AXIN1 and GSKβ to the plasma membrane, preventing the formation of the β-catenin degradational complex. 

Consequently, β-catenin translocates to the nucleus, where it binds to TCF and LEF and jointly regulates the transcription of downstream genes responsible for proliferation and cell survival [185]. WNT/β-catenin signaling is severely disrupted in most HCC cases. Somatic mutations account for about 40 to 60% of a disordered Wnt/β-catenin signaling [109]. The most prevalent mutations are in *CTNNB1*, which destabilizes and translocates β-catenin into the nucleus [78]. In addition, *CTNNB1* mutations decrease the expression of chemokines and subsequently the infiltration of immune cells (CD4^+^, CD8^+^, B cells, macrophages, and NK cells). This is thought to be a mechanism by which cancer cells escape the immune system [186]. Inactivating mutations in negative regulators *AXIN1*, *AXIN2*, and *APC* promote HCC [110]. In summary, *CNNTB1*, *AXIN1/2*, and *APC* mutations result in persistent aberrant activation of the Wnt/β-catenin pathway [109].

There is increasing evidence that non-mutational dysregulations also impair Wnt/β-catenin signaling and promote HCC. Recent data have shown that ubiquitin-conjugating enzyme E2 T (*UBE2T*), a direct binding partner and negative regulator of the E3 ubiquitin-ligase Mule, is significantly upregulated in HCC. Mule, in turn, is involved in the ubiquitination and degradation of β-catenin. Therefore, an abnormally high level of UBE2T results in reduced degradation and stabilization of β-catenin [111]. Furthermore, the transcription factor enhancer-binding protein 4 (TFAP4) has been reported to be upregulated and associated with an aggressive phenotype in HCC. Experiments demonstrated the activation of Wnt/β-catenin through direct binding of TFAP4 to the promoters of *DVL1* and *LEF1* [112]. Another factor significantly contributing to HCC progression is the DEAD-box RNA helicase DDX39. It is significantly upregulated and promotes cancer growth and metastasis through activation of Wnt/β-catenin. Analysis of the mechanism suggested an accumulation and translocation of β-catenin into the nucleus [113]. 

In the previous sections, we discussed the tremendous impact of RNA molecules on HCC development. Even in the context of Wnt/β-catenin signaling, circular RNA circ_0004018 negatively correlates with progressive liver cancer. Circ_0004018 interacts with miR-626 to eventually suppress Wnt/β-catenin signaling [114]. Hence, reduced levels of circ_0004018 reactivate Wnt/β-catenin and promote HCC. Another circular RNA exerting an antitumorigenic role is circ_0003418. Its downregulation dysregulates the Wnt/β-catenin pathway, making HCC cells more chemoresistant to cisplatin [115].

### 5.2. Receptor Tyrosine Kinase Pathway

The mitogen-activated protein kinase/extracellular signal-regulated kinase (MAPK/ERK) and phosphatidylinositol-three kinases (PI3K/Akt/mTOR) pathways are the best characterized and most frequently activated intracellular pathways in HCC [187]. They are activated by extracellular ligands (growth-factors) that bind to plasma membrane-anchored receptor tyrosine kinases such as IGFR, EGFR, PDGFR, VEGFR, FGFR, and HGFR/c-MET. MAPK/ERK signaling is constitutively activated in more than 50% of human HCC cases [188]. Constitutive activation of MAPK/ERK triggers the activation of the transcription factors cFos and AP-1/cJun which promote transcription of downstream genes driving proliferation, cell growth, de-differentiation, and cell survival [189,190,191]. Activation of PI3K through G-protein-coupled receptors (GPCR), IGFR, VEGFR, and others promotes the conversion of PIP2 to PIP3 and activates AKT and mTORC1. Uncontrolled activation of this pathway leads to carcinogenesis in most cases. Of note, mTORC1 and mTORC2 are upregulated in 40–50% of patients with HCC [116]. 

Most dysregulations are due to genetic or epigenetic alterations that affect the regulatory mechanism of signaling pathways. The Phosphatase and Tensin homologue (*PTEN*) is a potent tumor suppressor since it dephosphorylates PIP3 to PIP2 and attenuates PI3K signaling. In almost half of the cases, PTEN downregulation is associated with cancer progression in patients with HCC [117,118]. In contrast, upregulation of E3 ligase βTrcp and protein kinase CK1δ increases ubiquitination and degradation of Leucine Zipper tumor suppressor 2 (*LZTS2*). The depletion of LZTS2 leads to the progression and metastasis of HCC through activation of PI3K/AKT signaling [119]. Similarly, expression of the holiday junction recognition protein (*HJURP*) is increased in HCC and promotes cell proliferation by destabilizing p21via the MAPK/ERK and AKT/GSK3β pathway [120]. Chan et al. reported a recurrent inactivating *RSK2* mutation at 6.3% of HCC cohorts associated with a more aggressive tumor phenotype. Inactivation of RSK2 attenuates the SOS1/2-dependent negative feedback loop of MAPK/ERK signaling [121].

In addition to inactivating mutations, miRNAs and lncRNAs are also crucial regulators of gene expression. They are closely associated with the PI3K/AKT/mTOR pathway during oncogenesis [192,193]. For instance, oncogenic miRNAs such as miR155-5p, miR494, miR493, and miR519a are upregulated in various cancers. They induce cell invasion, migration, and proliferation. Moreover, these miRNAs inhibit apoptosis by directly targeting PTEN to prevent PIP3 to PIP2 reversion and promote activation of PI3K/AKT/mTOR [122]. Furthermore, Wu et al. comprehensively summarized 67 dysregulated PI3K pathway-related lncRNAs in HCC [123].

### 5.3. Vascular Endothelial Growth Factor and Further Signaling Pathways

The formation of new blood vessels from pre-existing vascular beds is a major characteristic of tumorigenic liver tissue [194]. Chronic liver injuries and inflammations cause fibrogenesis and impair the normal liver blood system. The disruption of blood supply leads to nutrition and oxygen shortage (hypoxia) [195]. Hypoxia in HCC, in turn, induces the release of specific growth factors such as hypoxia-inducible factors 1 and 2 (*HIF-1* and *-2*) and IGFs [171,196]. The release of angiogenic markers shifts the balance towards proangiogenic factors [194]. These include angiopoietins, angiogenin, FGF, HGF, Interleukin 4 and 8, PGF, PDGF, TGFβ, and VEGF [197]. However, a few other prognostic markers responsible for induction of angiogenesis have been identified in HCC. ERO1α, for example, is an endoplasmic reticulum resident oxidase that is significantly upregulated in tumorigenic tissue. It triggers the S1PR1/STAT3/VEGF-A signaling pathway, increasing the number of blood vessels and their density [124]. Moreover, imbalances in lncRNA expression can promote carcinogenesis. Increased lncRNA PAARH levels represses cellular apoptosis and promotes tumor growth and angiogenesis in HCC. Wei et al. demonstrated that PAARH physically binds to the HIF-1α subunit and thereby facilitates the recruitment of HIF-1α to the VEGF promoter [125]. These studies highlight the urgent need for new molecular targets to support the development of angiogenesis restricting therapies.

### 5.4. JAK/STAT Pathway

The JAK/STAT signaling pathway has a pivotal role in various cellular functions and can be activated by different cytokines and growth factors, such as interleukins, interferons, and EGFs. Each cytokine binds to its respective transmembrane receptors. Some of the intracellular receptor domains are associated with Janus kinases (JAKs), which get activated upon ligand-induced conformational change of the receptors. Activated JAKs phosphorylate Signal Transducers and Activators of Transcription (STAT). Activated STATs homodimers translocate into the cell nucleus and activate target genes such as *CCND1*, *BIRC5*, and *Mcl-1*. In HCC, the JAK/STAT signaling pathway is constitutively activated. Activation dysregulates expression of genes responsible for survival, angiogenesis, stemness, immune surveillance, invasion, and metastasis [198]. Constitutive activation of the signaling cascade can occur through several mechanisms. The most common example is an elevated interleukin-6 (IL-6) level that interacts with the receptor subunit GP130 and engages JAK phosphorylation and eventually STAT3 activation [126]. However, activating mutations e.g., in GP130, can lead to ligand independent STAT3 activation [127]. Somatic mutations in JAK1, IL6R, and IL6ST also result in constitutive JAK/STAT signaling [128].

### 5.5. Transforming Growth Factor-Beta Pathway

The transforming growth factor-β (TGF-β) signaling pathway regulates various physiological aspects in embryogenesis and adult tissue homeostasis [199]. The TGF-β family members are also involved in pathophysiological mechanisms that trigger severe diseases. TGF-β has a dual role of suppressing early stages of tumorigenesis but contributing to the migration and metastasis of HCC [129]. Liao et al. also proposed a mechanism of sustained TGF-β/SMAD signaling in HCC. Permanent activation is achieved by directly binding activated SMAD3 to protein tyrosine phosphatase receptor epsilon (*PTPRε*) promoters. Abundant PTPRε, in turn, facilitates the recruitment of SMAD3 to TGFBR1, resulting in a positive feedback loop [129]. 

The growth factor TGF-β is known to be the major contributor to the pro-metastatic nature of cancer. The transition from epithelial to mesenchymal cells is promoted through TGF-β induced SMAD2/SMAD3 activation. The activated proteins in turn operate as transcription factors promoting the transcription of genes involved in the dissolution of cell junctions and detachment from adjacent tissue [130]. Qu et al., also demonstrated intercellular communication of HCC cells by exosomes promoting their mutual invasion and migration. The exosomes decrease E-cadherin expression while activating TGF-β/SMAD signaling, which induces vimentin expression and, eventually, endothelial mesenchymal transition (EMT) [131]. In addition to its pro-metastatic properties, TGF-β also promotes immune surveillance escape in HCC [200].

## 6. Current Treatment Strategies

As mentioned above, therapeutic strategies always consider the HCC tumor status based BCLC criteria at baseline, the preserved liver function as well as tumor progression upon treatment or intolerance / adverse events of a current treatment (“treatment stage migration”). 

Surgical resection as well as local ablative therapies including radiofrequency ablation (RFA) and microwave ablation (MWA) are established therapeutic option in BCLC stage 0 and BCLC stage A, considering a preserved liver function of the individual patient. RFA in tumor nodules <2 cm offer same survival rates as surgical resection. RFA beyond 2 cm diameter is associated with lower rates of complete responses and higher recurrence rates. For patients with tumor nodules between 2–4 cm MWA is the suggested technique as it induces more intensive necrosis. 

Larger tumors benefit from surgical resection and should not be limited by tumor size alone as long there is no vascular invasion, or the prediction of the postoperative remnant liver function is assumed to be inadequate. Further, orthotopic liver transplantation is recommended for those patients inside MILAN criteria or for patients that fulfill the “extended criteria for liver transplantation” and responded to local therapy in terms of downsizing the tumor burden (see above).

### 6.1. Transarterial Chemoembolization (TACE) for BCLC Stage B

In intermediate-stage HCC, classified as BCLC stage B, transarterial chemoembolization (TACE) is a standard treatment. This approach is based on the European and American guidelines [10,23]. TACE can also be used for early-stage HCC (BCLC stage A) as a bridging therapy to liver transplantation, or to receive better respectability in neoadjuvant intentions [201,202]. 

In general, TACE is a locoregional therapy which is applicated by percutaneous access and deep cannulation of the hepatic artery system. Liver circulation is unique because of the dual blood supply by the portal vein and hepatic artery. In healthy liver tissue, the portal vein is responsible for 80% of the blood supply. The remaining 20% of blood supply is provided by the hepatic artery. In HCC, 99% of the blood supply to hepatic tumors is delivered by the hepatic artery [203,204].

The main goal of TACE is to enable the application of a chemotherapeutic agent and the additional embolization of the feeding vessel. Clinicians regularly differentiate between two techniques of embolization. The conventional TACE (cTACE), which uses lipiodol as a carrier agent, and DEB-TACE which uses drug-eluting beads (DEB). Worldwide, the most used chemotherapy agents for TACE for treating HCC, are doxorubicin, epirubicin, mitomycin C and cisplatin [205].

#### 6.1.1. Conventional TACE (cTACE)

Lipiodol was first used to treat HCC with labeled Iodine-131 [206]. If administered intra-arterially, the iodized oil lipiodol has shown to be selectively retained in tumor tissue. Iodine-131 is initially used as a contrast agent for radiographic diagnosing of HCC [207]. The mechanism of retention of lipiodol in and surround the tissue of HCC is not finally known. The effect is most likely due to the special arterial vascularization of HCC, which radiologically manifests itself as an early arterial contrast enhancement and a late washout [207]. It is discussed that lipiodol is absorbed by a membrane transporter in HCC cancer cells. After transfer into the intracellular matrix the transport mechanism is disabled by hypoxia, thus lipiodol retained within the hepatocellular tumor cells [208].

The goal of conventional TACE is hypoxic cell damage by injecting a mixture of lipiodol (iodized oil) and cytotoxic chemotherapies. The injection of the emulsion consisting of lipiodol, and chemotherapy is completed by the application of an embolic product (gelatin sponge particles or other solid embolic agents) to increase the cytotoxic effects [209]. The vascular occlusion provokes a hypoxic metabolic state in the tumor cell. Furthermore, the reduced vascularization increases the effectiveness of chemotherapy. The reason for this is presumably the delayed elimination of the cytotoxic substances. Further, the vascular occlusion leads to tumor necrosis by reduced arterial blood flow. On the other hand, vascular occlusion provokes formation of new blood vessels, for which reason the tumor tissue increases its own supply of oxygen and nutrients [210]. The patient’s outcome-benefits from additional vascular occlusion are yet not clear [211,212].

Meta-analyses have demonstrated superiority of cTACE over best supportive care for intermediate-stage HCC [213]. Nevertheless, the procedure of cTACE in means of the amount of the installed volume and concentration is not standardized, yet.

#### 6.1.2. Drug-Eluting-Bed TACE (DEB-TACE)

DEB-TACE is a procedure to deliver a combination of agents for embolization and cytotoxic chemotherapy (e.g., doxorubicin, epirubicin) at once. The chemotherapeutic drug is loaded into microspheres, from which it is released slowly to achieve sustained targeted release of cytotoxic agents. The application of microspheres leads to vascular occlusion. Nevertheless, microspheres cannot cross the system of blood capillaries. The effect of the cytotoxic component is due to drug-release in the distal arterioles. The further distribution follows by diffusion or convection [202]. The cytotoxic effect depends on microsphere size, loading degree, choice of chemotherapeutic agents and types of microspheres [214,215,216]. The systemic effect of chemotherapy is lower compared to cTACE due to the simultaneous embolization [217,218].

Randomized control trials reported no difference in tumor response and overall survival (OS) between cTACE and DEB-TACE However, DEB-TACE has been confirmed as the more standardized technique [217,219,220].

#### 6.1.3. Combination of TACE with Systemic Therapy

Patients with HCC in intermediate stages are typically treated with TACE. Nevertheless, TACE is regarded as a palliative treatment option. It is conceivable that the addition of effective systemic treatments may cause synergistic effects and may result in better tumor response rates and improved survival outcomes. 

TACE leads to hypoxic changes in HCC tissue which increases VEGF activity due to the upregulation of VEGF in the remaining tumor tissue [221]. Upregulation of VEGF, in theory, promotes tumor revascularization and local recurrence. Tyrosine kinase inhibitors (TKIs) can inhibit tyrosine kinases involved in the VEGF signaling pathway, reducing angiogenesis, and leading to delayed tumor recurrence [28]. TKIs can also inhibit RAF kinase, associated with inhibiting the MAPK/ERK pathway leading to reduced cell proliferation [221]. Furthermore, there are strong bidirectional relationships between angiogenesis and immunity [222,223].

Minimally invasive, locoregional procedures such as TACE or RFA activate the immune system to induce local inflammation and release of tumor antigens [224]. Therefore, some pathophysiological hypotheses have been investigated in clinical practice. In this regard, the administration of immune checkpoint inhibitors has been evaluated to activate immune cell response against the tumor antigens released by TACE. In a pathophysiological manner, it is discussed to prevent intrahepatic micro-metastases, a key factor of intrahepatic tumor recurrence [225].

Several clinical studies currently investigate the combination of TACE and immunotherapy and/or therapy with TKI. Although pathophysiological sound, prospective studies have failed to prove efficacy for this dual mechanism. Therefore, combination of TACE with TKI cannot be recommended (see below).

#### 6.1.4. Current Knowledge: Combination of TACE + Immunotherapy or TKI

There are some signals of additional or synergistic effects of TACE and systemic therapy. Signals for the efficacy of a combination of TACE and TKI are derived from the TACTICS trial. In this study, TACE plus sorafenib showed a prolonged progression-free survival (PFS) compared with TACE alone (25.2 months vs. 13.5 months) [226]. In contrast to PFS, the analysis of median OS as a primary aim of the study did not differ between TACE plus sorafenib and TACE alone (36.2 months vs. 30.8 months; hazard ratio 0.861; *p* = 0.40). Similarly, in a retrospective study of Lenvatinib and TACE, the combination showed improved clinical outcome compared with TACE alone [227].

Nevertheless, many randomized clinical trials failed to show any clinical benefit of combination therapy compared to TACE alone. SPACE trial and TACE-2 trial compared TACE plus sorafenib versus TACE alone. Both studies failed to demonstrate either a clinical or a benefit in OS. Further, BRISK-TA (TACE + brivanib) and ORIENTAL (TACE + orantinib) are also negative trials [228,229,230,231]. Based on these prospective randomized clinical studies, combination of TACE and systemic treatment is not recommended in international guidelines. 

There are several ongoing studies mainly with TACE plus immune-oncological targets. The combination of TACE and tremelimumab showed favorable outcomes, with a partial response rate of 26% and overall survival of 12.3 months (NCT01853618) [232]. A phase III trial of combination therapy with TACE plus durvalumab and/or bevacizumab (EMERALD-1 trial) is ongoing (NCT03937830). Double immune-checkpoint inhibition is investigated in an ongoing phase II trial of CTLA-4 /PD-L1 blockade using durvalumab and tremelimumab following TACE (NCT03638141). TACE-3 is a phase II/III randomized trial of Nivolumab in combination with TACE for patients with intermediate-stage HCC (NCT04268888) [233]. The phase I/II clinical trial PETAL evaluates the combination of TACE and pembrolizumab (NCT03397654) [234]. CheckMate74W is a phase III trial designed to investigate the combination of nivolumab plus ipilimumab in combination with TACE (NCT04340193). A further interesting study is the German phase II IMMUTACE (NCT03572582) which evaluates the effects of TACE in combination with nivolumab for intermediate-stage HCC. IMMUTACE met its primary endpoint (overall response rate, ORR) and demonstrated the efficacy of the combination therapy in the setting of intermediate-stage HCC. More data concerning the very important secondary endpoints like overall survival and progression-free survival must be awaited. Further randomized phase III clinical studies are necessary to evaluate the benefit of the combination of TACE and immunotherapy in the current therapeutic landscape. 

LEAP-012 is an ongoing randomized clinical phase III study to evaluate TACE with or without Lenvatinib plus Pembrolizumab for intermediate-stage HCC not amenable to curative treatment (NCT04246177) [235]. 

#### 6.1.5. Current Therapy Recommendations for TACE in HCC

TACE is widely indicated for patients with HCC in BCLC stage B. It has been recommended to use the more standardized technique of DEB-TACE. To date, there is no evidence from clinical trials for the combination of TACE and systemic therapeutic agents, so far.

Due to higher response rates of modern system therapy, clinical guidelines recommend early switch from TACE to systemic therapy regimen when the patient is refractory to TACE. In principle, treatment guidelines are recommended that the indication for the continuation of TACE should be reviewed in a multidisciplinary tumor board after two treatment cycles. Combination of TACE and systemic treatment cannot be recommended and patients who receive locoregional therapy should not be combined with systemic therapy outside of clinical studies.

### 6.2. Combination of TACE and RFA (Radiofrequency Ablation)

For patients with HCC with tumor diameters between 3 cm and 5 cm, a randomized clinical trial demonstrated that sequential therapy with TACE followed by RFA significantly improves overall survival (1-, 3-, and 5-year OS rates 94%, 69%, and 46%, respectively) compared to RFA alone (1-, 3-, and 5-year OS-rates 82%, 47%, and 36%, respectively) [236]. A clinical benefit for the sequence of TACE-RFA further indicated a study examining tumors smaller than 7 cm [237]. 

#### Current Therapy Recommendation for TACE and RFA in HCC

In patients with preserved liver function and low or moderate portal hypertension, chemoembolization prior to RFA or other thermal ablation therapy is recommended if the HCC lesion is >3 cm and <5 cm. TACE creates a preparatory reduced or complete devascularization of the tumor and its surroundings, which significantly increases the effect of a timely thermal ablation.

## 7. Systemic Therapy for HCC, BCLC C

### 7.1. Immunotherapeutic Strategies in HCC Treatment

#### 7.1.1. IO combination Therapies in 1st-Line Therapy of HCC

IO combination therapy has been introduced and practice changing in first-line therapy of HCC in BCLC stage C. The combination of atezolizumab (PD-L1-inhibitor) and bevacizumab (VEGF-inhibitor) showed encouraging antitumor activity and safety in a phase 1b trial involving patients with unresectable hepatocellular carcinoma [238]. Based on the results of the IMbrave-150 trial the combination of bevacizumab and atezolizumab became a new standard of care in first-line treatment of HCC in BCLC stage C [13,14]. In this randomized phase III clinical trial, the combination of atezolizumab and bevacizumab has demonstrated superiority compared to sorafenib. (Median OS 19.2 months vs. 13.4 months (*p* < 0.001)) [239] which leads to recommendation as first-line therapy in several international guidelines.

The combination of tremelimumab (CTLA-4 antagonist) plus durvalumab (PD-L1 antagonist) showed significantly improved median OS versus sorafenib as first-line therapy in a phase 3 trial (HIMALAYA). The median OS was 16.43 months with tremelimumab/durvalumab and 13.77 months with sorafenib [240]. Up to date, approval by FDA and EMA is awaited. 

Another study of nivolumab in combination with ipilimumab in patients with advanced hepatocellular carcinoma and no prior systemic therapy is still ongoing (CheckMate 9DW, NCT04039607). 

In the phase 3 LEAP-002 trial the combination of pembrolizumab and lenvatinib versus lenvatinib missed its primary endpoints [241]. It is important to note that lenvatinib in the control arm has performed much better than in the original RCT (median OS with Lenvatinib/pembrolizumab was 21.2 months vs. 19.0 months with Lenvatinib). In the approval study of Lenvatinib (REFLECT trial) the median OS was 13.6 months for lenvatinib. This effect could be explained by a better selection of patients, the better management of side effects, and the effect of better subsequent lines of systemic treatments after progression of the initial therapy.

An open-label randomized phase III trial (COSMIC-312) recently investigated the combination of cabozantinib plus atezolizumab versus sorafenib alone for the treatment of advanced hepatocellular carcinoma in the first-line setting. Median OS was 15.4 months with the combination cabozantinib plus atezolizumab and 15.5 months with sorafenib, no statistical significance was reached. Nevertheless, the study showed a benefit for PFS in the combination group (6.8 months vs. 4.2 months in the sorafenib group) [242].

The Rationale-301 study (phase III) demonstrated non-inferior median overall survival for tislelizumab (PD-1-inhibitor) versus sorafenib in patients with previously untreated unresectable HCC. Rationale-301 was designed as a non-inferiority study and met its primary endpoint of median OS (15.9 months versus 14.1 months) [243]. But the median PFS with tislelizumab was 2.2 months compared with 3.6 months with sorafenib. No approval has been signed either by FDA or by EMA, yet. 

A clinical phase II trial evaluated the efficacy of regorafenib plus tislelizumab as first-line therapy for patients with advanced HCC (NCT04183088). The phase II RENOBATE trial evaluates the combination of regorafenib and nivolumab in unresectable HCC (NCT04310709). The combination of regorafenib with camrelizumab or pembrolizumab is also subject of research (NCT05048017).

The clinical effects and benefits neither by other immunotherapeutic agents (nivolumab/ipilimumab or pembrolizumab) but also TKI therapy after failure of atezolizumab/bevacizumab in first-line therapy remains unclear. In this regard, further data from clinical studies will be required in the future.

#### 7.1.2. Clinical Data for Other IO-Based Treatment Regimens in Later Therapy Lines

Besides atezolizumab, several other immune-directed agents are currently on the market and under investigation in HCC therapy. Nivolumab as another checkpoint inhibitor binds to the PD-1 receptor on T cells and prevents interaction with the PD1 receptor ligand. Single-agent nivolumab showed durable responses and promising survival in patients with advanced hepatocellular carcinoma in the initial phase I/II dose escalation and expansion trial CheckMate-040 [244]. Based on this data, the FDA granted accelerated approval in the year 2017 for nivolumab for the treatment of HCC in patients who have been previously treated with sorafenib. The continued approval of the agent was planned to achieve by the phase 3 trial CheckMate-459 trial, but the study failed to meet its primary endpoint of overall survival [245]. Hence, FDA opposes the approval of nivolumab for second-line treatment in advanced HCC in 2021. In the CheckMate-040 trial, the study cohort investigating the combination of nivolumab plus ipilimumab showed a manageable safety profile with promising objective response and durable response rates. The regimen of arm A (4 doses of nivolumab 1 mg/kg plus ipilimumab 3 mg/kg every 3 weeks then nivolumab 240 mg every 2 weeks) received accelerated FDA approval for second-line treatment in the US based on the results of this study with an objective response rate of 32% [246]. Recently published data at a minimum follow-up of 44 months showed a median OS at 22.2 months for the approved dosing regimen [247]. Notably, besides FDA there is no positive approval in Europe by the EMA. 

The use of pembrolizumab (PD-1-inhibitor) as second-line therapy in patients with advanced HCC is investigated in KEYNOTE-224 trial. In the phase II trial, the major efficacy outcome measure was confirmed by overall response rate (assessed by independent central review (ICR). The confirmed ICR-assessed overall response rate was 17% (in 18 of 104 patients), with one complete response and 17 partial responses [248]. Because of these results, FDA granted accelerated approval to pembrolizumab for hepatocellular carcinoma for patients who have been previously treated with sorafenib. Updated efficacy data of KEYNOTE-224 reported in 2022 demonstrated objective response rates of 18.3% In the subsequent analysis, median progression-free survival was 4.9 months and median overall survival was 13.2 months [249]. Nevertheless, we have no approval from the EMA for pembrolizumab after progression with sorafenib treatment in HCC. A later study of pembrolizumab as second-line therapy in patients with advanced hepatocellular carcinoma (KEYNOTE-240; phase III) did not meet the combined primary endpoints of OS and PFS [250]. The FDA panel left the approval in place while waiting on results from KEYNOTE-394. KEYNOTE-394 compared pembrolizumab with best supportive care and offered significant improvements in overall survival, progression-free survival, and overall response rate compared with placebo and BSC. The median OS was 14.6 months for pembrolizumab and 13.0 months for placebo. PFS was 2.6 months for pembrolizumab compared with 2.3 months for placebo [251].

Based on these data, there is currently only insufficient clinical evidence for the use of immunotherapy in later lines of therapy (>second line). Therefore, the identification of patients’ subtypes based on molecular findings and pathophysiology is mandatory and clinically needed for tailored and individualized therapy in HCC.

### 7.2. Tyrosine Kinase Inhibitors

Tyrosine kinases contribute to the phosphorylation status of proteins in diverse signaling cascades. They often act as transmembrane receptors with an extracellular N-terminal domain for binding the ligand and an intracellular C-terminal domain for autophosphorylation of downstream proteins. In HCC the C-terminal domains of some receptor tyrosine kinases contain mutations that lead to constitutive activation. Inhibition of phosphorylation leads to antitumor and antiangiogenetic effects in HCC. In HCC therapy TKIs as multi-kinase inhibitors mainly target the VEGF receptor, PDGF receptor, RAF receptor, FGF receptor, KIT receptor, and RET receptor [252].

One point of pathophysiological attack is the inhibition of angiogenesis in the microenvironment of the tumor tissue as well as cell proliferation by TKIs [252,253]. Receptors involved in angiogenesis signaling are mainly VEGF receptor, PDGF receptor, and FGF receptor [254]. Several receptor tyrosine kinase-associated signaling pathways activated lead to tumor cell proliferation (WNT-β-catenin, RAS–MAPK, AKT–mTOR, EGFR, IGFR, HGF–MET) [255]. Furthermore, the activation of distinct EGF-receptor signaling pathways (especially the two pathways: Ras–Raf–MEK–ERK and PI3K–AKT–mTOR) are involved in the development of HCC and EGF receptor inhibition by TKI is a refined strategy to inhibit tumor growth [254].

#### 7.2.1. Sorafenib in 1st Line Therapy

Sorafenib is an orally administered multikinase inhibitor and the first approved targeted therapy for patients with HCC in BCLC C. Important target structures for sorafenib are the VEGF receptor, PDGF receptor, and RAF-kinase [252]. Raf kinase inhibition blocks the Raf signaling cascade (MAPK/ERK pathway) which leads to reduced cell division and proliferation [256]. By inhibiting signal transduction at the VEGF receptor (VEGFR-1, VEGFR-2, VEGFR-3) and PDGF receptor, which also use the Raf signaling cascade, tumor angiogenesis and cell proliferation are inhibited [257,258]. 

Sorafenib is approved by the Food and Drug Administration (FDA) for the treatment of unresectable HCC. In the controlled phase III approval study SHARP sorafenib showed a significant prolongation of median OS of 10.7 months compared with 7.9 months in the placebo group (hazard ratio 0.69) [221]. Another phase III study in the Asia-Pacific region (ORIENTAL) showed similar results with a median OS of 6.5 months for sorafenib compared with 4.2 months in the placebo group (hazard ratio 0.68) [10]. Importantly, both studies only included patients with preserved liver function (Child-Pugh A, 5–6 points). The patients, therefore, had a well-compensated liver function. In the GIDEON study, a prospective register of 3202 patients, the investigators demonstrated the safe use of sorafenib in patients with more advanced liver cirrhosis (Child-Pugh B, 6–8 points) [259]. 

In some of the latest studies, evaluating immunotherapy as a first-line setting, sorafenib was the active comparator. In these studies, the median OS of sorafenib monotherapy as first-line treatment significantly improved up to 14.7 months (CheckMate-459) [245]. This may be explained by better management of side effects and advances in healthcare with better second-line and later-line therapies. 

Sorafenib was also investigated in the adjuvant setting after curative intended resection of HCC, but the phase III study STORM failed to reach its primary endpoint [260]. In the TACTICS trial sorafenib was found to be effective as a concurrent treatment of TACE with a prolonged PFS but no significant difference in median OS [226]. 

In summary, sorafenib is indicated for the treatment of patients with unresectable HCC in BCLC C or progression after TACE. Clear benefit has been demonstrated for patients with compensated liver function (Child-Pugh A, 5–6 points). 

#### 7.2.2. Lenvatinib in 1st-Line Therapy

Lenvatinib was the second TKI approved for therapy of advanced HCC. It acts as a multitarget TKI as well, which inhibits VEGF receptor, PDGF receptor, FGFR receptor, KIT, and RET [1]. Targeting the FGF signaling pathway in HCC differentiates lenvatinib from sorafenib. Therefore, lenvatinib showed antitumor, antiproliferative, and antiangiogenic properties [261,262]. 

The approval of lenvatinib is based on a non-inferiority study. In the multicenter phase III REFLECT trial the investigators demonstrated that Lenvatinib was non-inferior compared to sorafenib concerning median OS. Lenvatinib reached a median OS of 13.6 months compared to 12.3 months in the sorafenib group (hazard ratio 0.92) [263]. Furthermore, the objective response rates improved to 24.1% with lenvatinib compared to 9.2% with sorafenib. Furthermore, a subgroup analysis of the REFLECT trial demonstrated a further improved median OS in those patients who initially responds to lenvatinib (median OS 22.4 months vs. 11.4 months; hazard ratio 0.61) [263]. Based on this data, lenvatinib is approved only for first-line treatment of unresectable HCC. 

Recently, the randomized, double-blind phase III trial LEAP 002 evaluated the efficacy and safety of Lenvatinib plus Pembrolizumab compared with Lenvatinib plus placebo for first-line treatment of unresectable HCC BCLC C. Despite failing the primary endpoints of the study (combined endpoint of median OS and PFS), the trial still demonstrated an impressive median OS of 19.0 months for lenvatinib monotherapy [241]. These data lead to discussion, about if lenvatinib as first-line therapy in case of contraindications for atezolizumab/bevacizumab should be the standard therapy as first-line TKI. 

In summary, lenvatinib is indicated for the treatment of treatment-naive patients with unresectable HCC BCLC C, probably if therapy with atezolizumab/bevacizumab is not possible. There are experimental data from subgroup analyses that support the hypothesis of patients with HCC based on NAFLD/NASH are performing better under therapy with lenvatinib [264,265]. Prospective data on this issue are missing but this fact underlies the importance of a better understanding of molecular HCC pathogenesis for tailored and individualized medical treatment. 

#### 7.2.3. Regorafenib in 2nd Line Therapy after Sorafenib

Regorafenib is an inhibitor of VEGF receptor, PDGF receptor, FGF receptor, RAF, RET, KIT [252]. Regorafenib is approved based on the results of the randomized, phase III RESORCE trial. The treatment group with regorafenib reached a statistically significant extension of the median OS compared to placebo (10.6 months vs. 7.8 months, hazard ratio 0.63) [266]. Regorafenib is approved for the treatment of HCC who have been previously treated and responded to sorafenib. 

Since the combination of atezolizumab and bevacizumab is the new first-line standard, the use of regorafenib after immunotherapy as second-line therapy must be evaluated. This question is addressed in another phase II clinical trial (NCT05134532).

#### 7.2.4. Cabozantinib in 2nd Line Therapy

Cabozantinib inhibits the VEGF receptor (VEGFR-1, VEGFR-2, VEGFR-3), RET and KIT. Cabozantinib also inhibits MET and AXL, which have previously been associated with resistance to sorafenib [252]. Cabozantinib is approved for the treatment after treatment with sorafenib in patients with unresectable HCC BCLC C. The approval is based on the randomized phase III CELESTIAL trial. 707 patients were randomized to receive cabozantinib or placebo (ratio 2:1). Cabozantinib reached a statistically significant benefit in a median OS of 10.62 months compared to 8.0 months in the placebo group [242]. The recommended dose of cabozantinib is 60 mg once daily. Nevertheless, a dose reduction to 40 mg or 20 mg per day to reduce side effects are a possible treatment option. 

Furthermore, there are phase II studies evaluate cabozantinib to treat recurrent HCC after liver transplantation (NCT04204850) or as second- or third-line treatment in HCC patients that progress on or are intolerant to immune checkpoint inhibitors, including anti-PD-1 and anti-PD-L1 antibodies (NCT04435977).

#### 7.2.5. Ramucirumab in 2nd Line Therapy in AFP-Overexpressing HCC

Ramucirumab is a monoclonal antibody and binds to the VEGF receptor 2. Binding of ramucirumab to the extracellular domain of the VEGF R2 receptor prevents binding of the ligands VEGF-A, VEGF-C, and VEGF-D, which consecutive prevents tumor angiogenesis. Ramucirumab is approved for second-line treatment of patients with AFP-overexpressing tumors (>400 ng/mL). The approval is based on the results of the randomized phase III clinical trial REACH-2. The antibody significantly increased median OS (8.5 months vs. 7.3 months, hazard ratio 0.71) and PFS (2.8 months vs. 1.6 months, hazard ratio 0.452) compared to the placebo group [267]. 

## 8. Future Directions

### 8.1. Immunotherapeutic Strategies

Immune checkpoint inhibition as an antibody-based therapy that blocks receptor-ligand interactions between effector lymphocytes and cancer cells, which normally would have negative regulatory effects on the tumor-directed cytotoxic immune response. 

The immune system is the largest natural counterplayer for external invading pathogens and cancerous development. Especially T-cells develop an anti-tumoral response by expressing specific receptors for tumor antigens [268]. In general, carcinogenesis drives genomic mutations occasionally leading to amino acid changes in proteins and consequently to cancer neoantigens [269]. To prevent immune overreactions, effector lymphocytes express immune checkpoints that represent receptor proteins exposed on the cell surface. Several tumors, including HCC, exploit this circumstance by producing corresponding ligands. Upon binding, effector lymphocytes are markedly restricted in their activity. Such co-inhibitory receptors include cytotoxic T-lymphocyte-associated antigen 4 (CTLA4), programmed cell death protein 1 (PD-1), T-cell immunoglobulin and mucin domain containing-3 (TIM-3), lymphocyte activation gene 3 (LAG3) and others [270]. Since the liver is daily confronted with exogenous molecules and metabolites, it has built up an anti-inflammatory environment by nature [271]. Hepatic non-parenchymal cells (NPCs) which are composed of Kupffer cells, liver sinusoidal endothelial cells (LSECs), and stellate cells mainly contribute to the maintenance of this tolerogenic milieu. Kupffer cells as the liver resident macrophages generate immune inhibitory molecules and enzymes such as IL-10, prostaglandins, and indoleamine 2,3-dioxygenase-1 (IDO) [272]. LSECs produce high levels of PD-L1 and additionally promote induction of T regulatory cells (Tregs) via TGF-β production. Hepatic stellate cells are responsible for HGF production which in turn promotes Treg accumulation [273]. Given the large anti-inflammatory properties of the liver, ICI represents a new and highly efficient way to treat tumor cells, especially in immune-tolerant tissue. 

To date, multi-tyrosine kinase inhibitors (TKI) such as Lenvatinib and sorafenib are regarded as first-line therapies for patients with unresectable HCC [10,274]. However, recent guidelines recommend considering immune checkpoint inhibitors [274]. Recent studies have shown that combinational long-term therapies with immune checkpoint- and growth factor inhibition are superior to conventional TKI therapies in terms of overall survival and life-year gain. In this context, Finn et al. demonstrated the superior effects of atezolizumab (anti-PD-L1 humanized monoclonal antibody) and bevacizumab (anti-VEGF humanized monoclonal antibody) combination for unresectable HCC [239]. Other immune checkpoint inhibitors such as nivolumab and pembrolizumab (anti-PD-1 humanized monoclonal antibodies) are also considered monotherapies [274]. However, these therapeutics failed to prolong overall survival compared to lenvatinib or sorafenib but showed a non-significant trend with improved long-term survival rates [275]. On this occasion, a double-blinded phase III clinical trial showed that 20% of the patients receiving pembrolizumab remained free from HCC progression for more than a year. In the control group, only 7% remained progression-free [250]. Even after the failure of ICI-based first-line therapy, lenvatinib or sorafenib are applicable as continuing second-line therapy. A study conducted by Cabibbo et al. demonstrated the beneficial effects of first-line combinational treatment with atezolizumab and bevacizumab followed by lenvatinib or sorafenib concerning overall survival (24 months) and life year gain (0.5 years) [276]. Nowadays, immune checkpoint inhibitors have found their way into clinics as an invaluable treatment option for solid tumors. While today’s ICI is still considered for adjuvant and neo-adjuvant therapy, further clinical studies will verify the need for immune checkpoint inhibitors as the main therapeutic for tomorrow’s medicine against the progression of HCC.

### 8.2. Adoptive Cell Transfer

Adoptive cell transfer (ACT) is a form of therapy in which effector cells (e.g., lymphocytes) are sensitized, propagated ex vivo, and subsequently reinfused into the patient [277]. The most used lymphocytes for ACT are genetically recoded T cells that specifically recognize and target tumor cells. The two main strategies are either transgenic tumor antigen-specific T cell receptors (TCRs) or chimeric antigen receptors (CARs). Patients undergoing ACT treatment are usually preconditioned with cycloheximide and fludarabine to induce lymphodepletion. This treatment supports in vivo expansion of adopted and retransferred immune cells. CAR T cells have shown very promising results in the treatment of hematological malignancy but are still in development for solid tumors [278]. The tumor-specific antigen Glypican-3 (*GPC3*) is the most attractive target for CAR T cell-based treatment in HCC [279]. Clinical trials are currently underway to ensure the safety and efficacy of GPC3-directed CAR T cells (NCT04121273) [280].

### 8.3. Therapeutic Vaccination

The use of therapeutic vaccines against tumors primarily enhances the potency of specific anti-tumoral responses [278]. These responses are achieved by de novo priming of T cells directed against antigens expressed by HCC cells that enhance immune responses or expand the repertoire of HCC-specific responses [281]. Studies have shown that sole therapies should be avoided, but combination strategies with immune checkpoint inhibitors ACT should be considered [282]. While GPC3- and telomerase-based vaccines have not provided meaningful clinical results, tumor unique peptide identification through HLA-peptidomics is the more robust and personalized approach for anti-HCC vaccine treatment [283,284]. 

### 8.4. Locoregional Therapy Enhancement

Locoregional therapy enhancement describes a novel therapeutic approach in which the cancer tissue is presented as infected or stressed cells, which could generally enhance the activity of the cytotoxic immune system [278]. Two critical mechanisms can make a tumor more susceptible to the immune system. These include immunogenic cell death [285] and pathogen-related molecular triggers [286]. For example, oncolytic virotherapy uses viruses that selectively replicate in cancer cells to destroy them during their replicative and lytic cycle. Studies of oncolytic viruses revealed high potential efficacies in HCC cells. However, anti-tumoral effects are due to pathogen-induced activation of the immune system and not to oncolytic effects of the viruses as previously assumed [287,288]. Combined treatment with system therapies, such as check-point-based immune therapies, could potentiate the efficacy of locoregional treatments in HCC [278] but has failed in prospective clinical studies, so far (see above).

### 8.5. Novel Therapeutic Targets in HCC-Related Signaling Pathways

As discussed in previous sections, dysregulation of signaling pathways controlling many cellular processes like cell growth, proliferation, differentiation, invasion, metastasis, and cell viability can contribute significantly to the initiation and progression of HCC. Therefore, it is of utmost importance to identify therapeutic targets within HCC-promoting signaling pathways to restrict spreading and ultimately reduce tumor size. 

Unfortunately, most tumorous liver tissues, especially HCC, seem resistant to conventional chemotherapy and radiotherapy. The poor efficacy of anti-tumoral agents is partly due to the inefficient drug delivery and metabolism exerted by the cirrhotic liver. Metformin (METF) is a novel approach in chemotherapy that may improve prognosis in patients. METF significantly decreases cell growth by raising KLF6/p21 levels. Also, a reduction of prevalent HCC-tumor markers like *CK19* and *OPN* in HepG2 cells was achieved through METF treatment [289]. Another strategy for restricting tumoral growth involves the inhibition of hypoxia-induced angiogenesis via the ERK/HIF-1α/VEGF axis. Han et al. identified the phenazine biosynthesis-like domain-containing protein (*PBLD*) as the responsible tumor-suppressive marker. Besides downregulating HIF-1α/VEGF expression in HCC cells, PBLD also blocks VEGFR2 in endothelial cells [290].

The metastatic behavior of HCC cells is provoked by hypoxia and the aberrant activation of the TGF-β pathway. Consequently, cancer cells experience the transition from an epithelial to a mesenchymal state. The natural benzophenanthridine alkaloid sanguinarine impairs the proliferation and colony formation of HCC cells. Mechanical studies revealed the inhibitory effect of sanguinarine on HIF-1α and TGF-β signaling, which lowers the expression of EMT markers [291]. Further, histone deacetylase 6 (*HDAC6*) overexpression decreases β-catenin to attenuate canonical WNT/β-catenin signaling. HDAC6 inhibits EMT by increasing E-cadherin levels and decreasing pro-EMT factors such as N-cadherin, vimentin, and matrix metalloprotease-9 levels [292]. Recent studies have indicated the potential of resminostat/sorafenib combinational treatment in HCC. Exploration of the molecular mechanism revealed a reduction of platelet induced CD44 expression and deregulation of several genes involved in EMT. Furthermore, this specific drug combination reduced the phosphorylated ERK level, leading to inhibition of the MEK/ERK signaling pathway [293].

### 8.6. Non-Coding RNA-Based Therapies

RNA therapeutics are becoming increasingly important in the treatment of tumors. They show various advantages as they harbor low immunogenicity, are plasma membrane and nuclear membrane permeable, are chemically modifiable to increase stability, and can target any desired gene or/and gene product [294]. In this context, there are numerous options for targeting non-coding RNAs to repress the progression of HCC. To name a few, small interfering RNAs (siRNAs) can target and degrade specific lncRNAs through the RISC complex [295]. Similarly, therapeutic lncRNAs can inhibit the prooncogenic effects of endogenous overexpressed lncRNAs [296]. Another strategy involves the use of small-molecule inhibitors that mask the binding sites for lncRNA and prevent the association with other binding partners [297].

For example, the lncRNA HOTAIR recruits EZH2 to EMT-promoting sites to bind and simultaneously repress SNAIL [298]. Identifying the SNAIL-binding domain through bioinformatic fragmentation of HOTAIR [299] led to the development of a therapeutical RNA molecule, namely HOTAIR-said, which contains the SNAIL-binding domain and lacks EZH2-binding capacity. Thereby, HOTAIR-said physically blocks EZH2-HOTAIR-SNAIL complex formation [300]. These studies elucidate the enormous potential of non-coding RNA-based strategies that could drastically improve the prognosis of patients with HCC.

## 9. Conclusions 

The BCLC classification has been the standard for classification and stage-based therapeutic algorithms in the clinical management of HCC for several years. The therapeutic options include surgical resection, local ablative therapies, transarterial chemoembolization, and radioembolization for the intermediate stages (BCLC stage B). Liver transplantation is considered for tumor nodules characterized inside the extended MILAN criteria for the early stages. Clinical decision-making should base on a multidisciplinary board discussion incorporating all disciplines and local expertise.

For BCLC stage C patients, systemic therapy is recommended with growing therapeutic options and targets. Unfortunately, there has not been a single predictive biomarker (except for elevated serum AFP for ramucirumab) linked to the therapeutic response to any therapeutic agent. 

These circumstances emphasize the need for more translational studies and indicate the unmet need for further molecular driver-targeted therapies. We summarized the molecular pathogenesis of HCC regarding current “druggable” treatment options for systemic HCC treatment. 

Further and additional efforts are needed in the future, based on human histopathological tumor specimen, that must be obtained not only for the diagnosis but, especially to tailor personalized medicine in HCC.

## Figures and Tables

**Figure 1 biomedicines-10-03202-f001:**
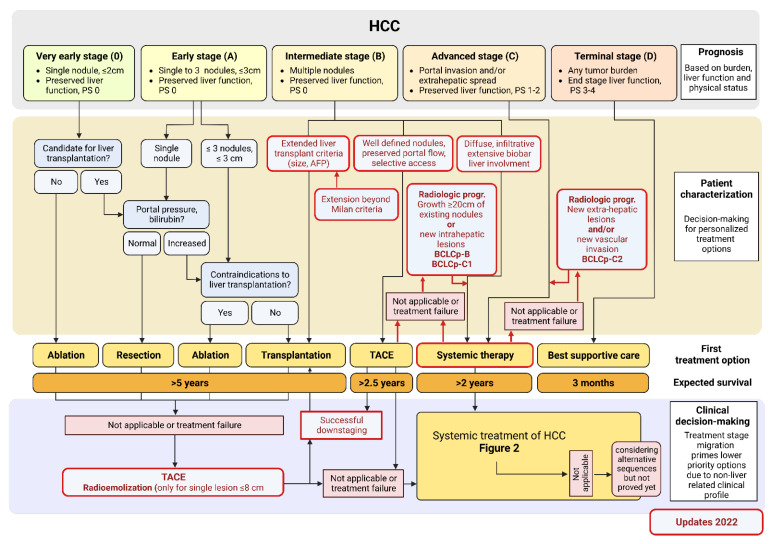
The BCLC classification system offers treatment recommendations based on tumor stage and liver function of patients with HCC. Treatment options are adjusted to the 5 stages of HCC (from very early to terminal stage). The prognosis is based on the tumor burden, liver function and physical performance status of the patient. Besides the recommended treatments, the BCLC system also includes prognosis and the expected survival time. Updated classifications and treatment options are illustrated as red boxes and arrows. This updated BCLC system further contains new workflows during radiological progression within between the stages (BCLCp-B, BCLCp-C1, BCLCp-C2) [13,14].

**Figure 2 biomedicines-10-03202-f002:**
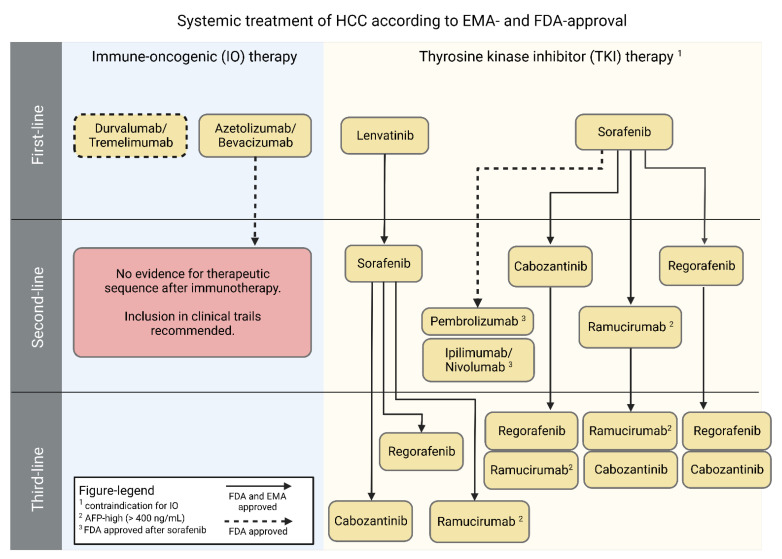
Algorithm of systemic therapy in HCC (BCLC stage C) based on approval by EMA and FDA. Scientific data supporting a therapy algorithm after 1st line therapy with Atezolizumab/ Bevacizumab is missing. Sorafenib is approved in the therapy of HCC regardless any other / previous therapy. Cabozantinib, ramucirumab and regorafenib are approved after previous therapy with sorafenib; Regorafenib for patients that has previously responded to sorafenib; Ramucirumab in patients with AFP-overexpressing tumors (AFP > 400 ng/mL).

**Figure 3 biomedicines-10-03202-f003:**
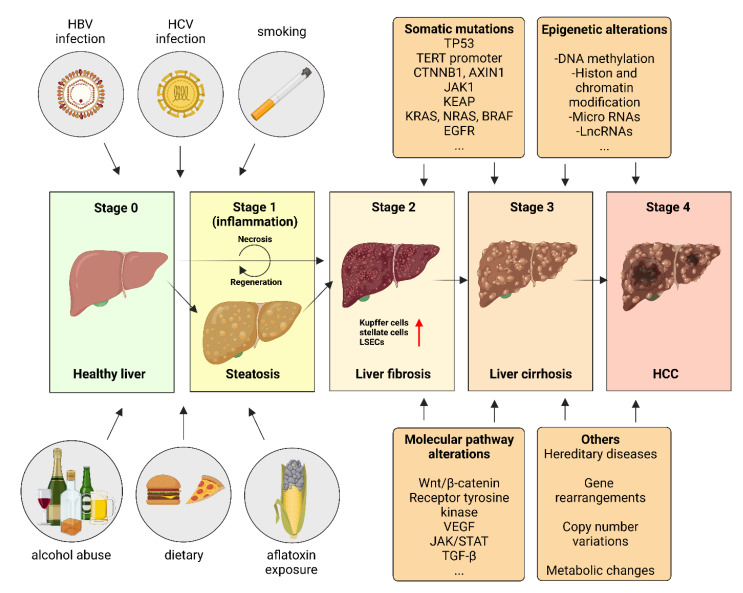
Etiology and risk factors promoting HCC carcinogenesis. Continuous chronic exposure to hepatic injuries caused by environmental factors (viral hepatitis, alcohol abuse, NASH, exposure to toxins, etc.) repeatedly damage the hepatocytes and induce inflammation eventually leading to liver cirrhosis. In this stage, the liver is susceptible to genomic instabilities. That is why it is more likely for the hepatocytes to accumulate somatic mutations, epigenetic changes, and gene rearrangements which can lead to metabolic changes and molecular pathway alterations. These dysregulations drive tumor progression and metastasis.

**Figure 4 biomedicines-10-03202-f004:**
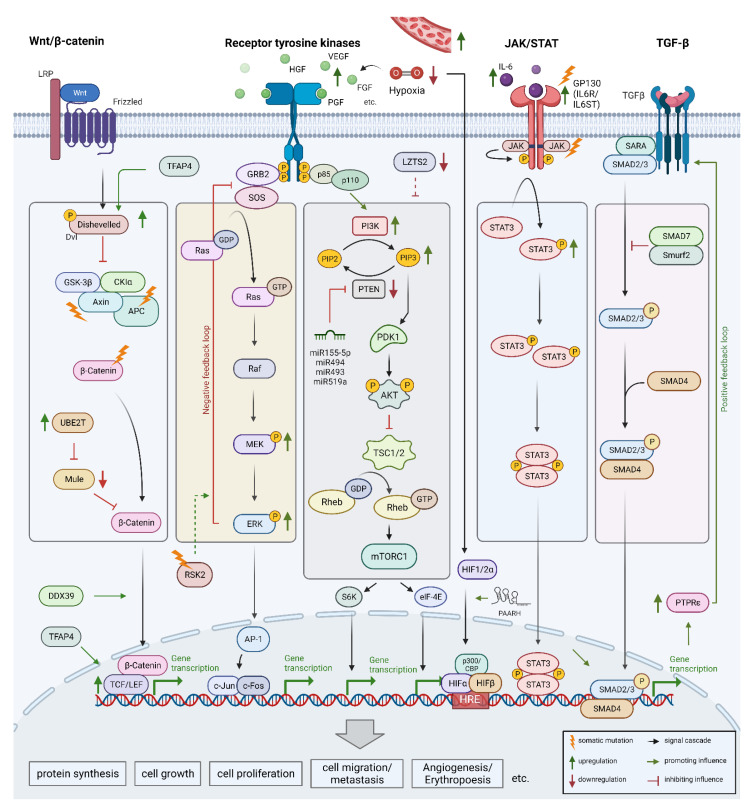
Most likely dysregulated key-signaling pathways are in HCC carcinogenesis. The occurrence of distinct somatic mutations and the influence of either inhibiting or activating factors all favor the constitutive activation of significant signaling pathways. Consequently, the cell evades any regulatory mechanism and strongly promotes tumor-typical properties including cell growth, cell proliferation, migration, and metastasis. Thus, smallest changes in the finely tuned signaling cascades result in hepatocarcinogenesis.

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
