# Peer review of "Pathogenesis and Current Treatment Strategies of Hepatocellular Carcinoma"

_biomedicines, 2022, doi:10.3390/biomedicines10123202_

Round 1

Reviewer 1 Report

This is a very nice, comprehensive, well structured and relevant review by Tumen et al. - most of all, it brings new valuable insights in the field of HCC, so well done to the authors. Moreover, the schematic representations done by the authors are extremely relevant. As a reviewer, it's all the time a real pleasure for me to revise such a manuscript.

Here are my comments about this elegant work to complete and strenghthen this work, only minors:

- Apoptosis is mentioned several times and especially its link to the P53 tumor suppressor. What about ferroptosis ? Please add 1-2 sentence about this new type of cell death which also seems to be important in the pathogenesis/maintenance of HCC. To do so, please refer to:

-Zhang X, Zheng Q, Yue X, Yuan Z, Ling J, Yuan Y, Liang Y, Sun A, Liu Y, Li H, Xu K, He F, Wang J, Wu J, Zhao C, Tian C. ZNF498 promotes hepatocellular carcinogenesis by suppressing p53-mediated apoptosis and ferroptosis via the attenuation of p53 Ser46 phosphorylation. J Exp Clin Cancer Res. 2022 Feb 28;41(1):79. doi: 10.1186/s13046-022-02288-3. PMID: 35227287; PMCID: PMC8883630.

- After: "PGK1, in turn, is an essential enzyme in the aerobic glycolysis pathway.", authors should discuss very briefly about the role hold by the Warburg effect in HCC tumorigenicity since it's a crucial element. It would then in my opinion complete this already nice work. To do so, please refer to 

-Bao MH, Wong CC. Hypoxia, Metabolic Reprogramming, and Drug Resistance in Liver Cancer. Cells. 2021 Jul 6;10(7):1715. doi: 10.3390/cells10071715. PMID: 34359884; PMCID: PMC8304710.

-Cassim S, Raymond VA, Dehbidi-Assadzadeh L, Lapierre P, Bilodeau M. Metabolic reprogramming enables hepatocarcinoma cells to efficiently adapt and survive to a nutrient-restricted microenvironment. Cell Cycle. 2018;17(7):903-916. doi: 10.1080/15384101.2018.1460023. Epub 2018 May 21. PMID: 29633904; PMCID: PMC6056217.

-Feng J, Li J, Wu L, Yu Q, Ji J, Wu J, Dai W, Guo C. Emerging roles and the regulation of aerobic glycolysis in hepatocellular carcinoma. J Exp Clin Cancer Res. 2020 Jul 6;39(1):126. doi: 10.1186/s13046-020-01629-4. PMID: 32631382; PMCID: PMC7336654.

Author Response

A point to point response has been attached.

Reviewer 2 Report

The authors present a narrative review on pathogenesis and treatment options for patients with HCC. This is an indeed an exhaustive review which however bring little new information in the field.

Major points:

- I'm a bit concerned about the term "counteracting" in the title and not sure it will help the readership understand what the article really is about. 

- The authors do not mention at all the BCLC 2022 update. Thus their work needs to be updated in terms of figures and strategies. I believe it is indispensable to discuss the treatment stage migration theme as well.

- Figures need to be reordered. Figure 2 (4?) is unreadable.

- The authors mention almost no information on hepatic resection and liver transplantation. Do these treatment options, which are in fact curative not fall under the nover term "counteracting"?

- The authors need to stress out that the BCLC treatment algorythm relates to cirrhotic patients alone!

- Several inaccuracies throughout the manuscript need to be seen again by the authors ie. in section 6 they quote "transplantation is preserved for those patients inside MI-LAN criteria." and then in the conclusion they quote "Liver transplan-tation is considered for tumor nodules characterized inside the extended MILAN criteria"

Minor point"

- English language needs revision throughout the text for grammatical errors.

Author Response

A point-to-point response has been attached.

Round 2

Reviewer 2 Report

The authors have adequately addressed my remarks.
